# Importance sampling scheme
# for the stochastic simulation of quantum spin dynamics

Stefano De Nicola

IST Austria, Am Campus 1, 3400 Klosterneuburg, Austria

## Abstract

The numerical simulation of dynamical phenomena in interacting quantum systems is a notoriously hard problem. Although a number of promising numerical methods exist, they often have limited applicability due to the growth of entanglement or the presence of the so-called sign problem. In this work, we develop an importance sampling scheme for the simulation of quantum spin dynamics, building on a recent approach mapping quantum spin systems to classical stochastic processes. The importance sampling scheme is based on identifying the classical trajectory that yields the largest contribution to a given quantum observable. An exact transformation is then carried out to preferentially sample trajectories that are close to the dominant one. We demonstrate that this approach is capable of reducing the temporal growth of fluctuations in the stochastic quantities, thus extending the range of accessible times and system sizes compared to direct sampling. We discuss advantages and limitations of the proposed approach, outlining directions for further developments.

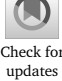

# 1 Introduction

Experimental breakthroughs in the simulation of isolated many-body quantum systems [1,2] have led to great theoretical interest in their far-from-equilibrium dynamics [3]. Concepts such as the thermalization of isolated quantum systems [4,5], or the absence thereof [6,7], and the discovery of novel non-equilibrium phenomena [8–10], have been the subject of intense experimental and theoretical exploration. Great progress has been achieved for one-dimensional systems, which include analytically solvable integrable models [11] and are often amenable to efficient numerical treatment via tensor-network based approaches [12–14]. However, the limitations of existing techniques call for the development of additional analytical and numerical tools to describe non-equilibrium quantum dynamics. This is particularly important in higher-dimensional settings, where no exact solutions are generally available and the applicability of tensor network methods is limited [15,16]. A number of directions are currently being explored, including linked cluster expansions [17] and neural network approaches [18–20].

An alternative technique, recently applied to many-body quantum spin systems, consists in exactly mapping unitary quantum dynamics to an ensemble of classical stochastic processes [21–25]. This approach is based on the disentanglement formalism [21–23,26], which provides an exact functional integral representation of the time-evolution operator. In this formalism, interactions are decoupled by means of Hubbard-Stratonovich (HS) transformations [27,28], and the resulting single-spin dynamics is then parameterized in terms of a set of classical *disentangling variables*, defined by a Lie-algebraic transformation [21–23,26,29,30]. Quantum expectation values are then obtained as classical averages over ensembles of stochastic trajectories of the disentangling variables. Early investigations have shown that the stochastic approach is immediately applicable to higher-dimensional settings, but its practical performance is limited by the exponential growth of fluctuations in the stochastic quantities as a function of time [24]. In recent work, the disentanglement formalism was applied in imaginary time, providing an analytical and numerical framework to study the ground states of quantum spin systems [31]. In this context, the identification of the saddle point trajectory of a suitable action, which provides the dominant contribution to observables, was used to perform an exact measure transformation. This resulted in an importance sampling scheme which greatly improves the performance of the numerical stochastic approach.

In this manuscript, we generalize the importance sampling scheme to real-time evolution. We begin by briefly recapping the main aspects of the disentanglement formalism in Section 2. The real-time importance sampling scheme in then introduced in Section 3, emphasizing similarities and differences with the imaginary-time case. The approach is first applied to local observables in Section 4, comparing it to direct sampling and discussing the role of fluctuations. Return probabilities are then considered in Section 5 in the context of dynamical quantum phase transitions [8,32]. We conclude in Section 6, summarizing our findings and discussing directions for further research.

# 2 Disentanglement formalism

The dynamics of a quantum state $|\psi_0\rangle$ under the action of a Hamiltonian $\hat{H}$ is encoded in the time-evolution operator $\hat{U}(t) \equiv \mathbb{T}\exp[-i\int_0^t \hat{H}(t')dt']$, where we set $\hbar = 1$ and $\mathbb{T}$ denotes time ordering: $|\psi(t)\rangle = \hat{U}(t)|\psi_0\rangle$. We consider a generic quadratic spin Hamiltonian

$$\hat{H} = -J\sum_{jkab}\mathcal{J}_{jk}^{ab}\hat{S}_j^a\hat{S}_k^b - \sum_{ja}h_j^a\hat{S}_j^a\,, \tag{1}$$

where the indices $j, k$ correspond to lattice sites and $a, b$, run over the generators of SU(2). The external fields $h_j^a$ and the interaction matrix $\mathcal{J}_{jk}^{ab}$ can in general be time-dependent and no specific boundary conditions are assumed. The constant $J$ is an overall coupling strength. The time-evolution operator corresponding to the Hamiltonian (1) is represented by a matrix whose size grows exponentially with the system size $N$, forbidding its direct evaluation for large systems. However, this issue can be circumvented by means of the *disentanglement formalism* [21–24, 26, 31]. In this approach, $\hat{U}$ is exactly written as a functional integral featuring matrices whose size is determined by the local Hilbert space dimension, e.g. they are $2 \times 2$ matrices for spin-1/2 systems. Below we recap the main features of this approach; additional details can be found in Ref. [24]. For simplicity, we consider systems initialized in a product state $|\psi_0\rangle = \otimes_i |\psi_0\rangle_i$; more general initial conditions can also be treated within the same formalism, e.g. by first performing imaginary-time evolution from a product state and subsequently evolving in real time. The time-evolution operator $\hat{U}$ admits the exact functional integral representation [21–23]

$$\hat{U}(t) = \int \mathcal{D}\varphi e^{-S_0[\varphi]} \prod_j e^{\xi_j^+(t)\hat{S}_j^+} e^{\xi_j^z(t)\hat{S}_j^z} e^{\xi_j^-(t)\hat{S}_j^-} , \tag{2}$$

where the *noise action* $S_0$ is given by

$$S_0[\varphi] \equiv \frac{iJ}{4} \int_0^t dt' \sum_{abjk} (\mathcal{J}^{-1})_{jk}^{ab} \varphi_j^a(t') \varphi_k^b(t') \tag{3}$$

and the *disentangling variables* $\xi_j^a(t)$ satisfy the equations of motion [22]

$$-i\dot{\xi}_j^+ = \Phi_j^+ + \Phi_j^z \xi_j^+ - \Phi_i^- \xi_j^{+2} , \tag{4a}$$

$$-i\dot{\xi}_j^z = \Phi_j^z - 2\Phi_j^- \xi_j^+ , \tag{4b}$$

$$-i\dot{\xi}_j^- = \Phi_j^- \exp \xi_j^z , \tag{4c}$$

with $\Phi_j^a \equiv h_j^a + J\varphi_j^a$ and initial conditions $\xi_j^a = 0$. The system of equations (4) encodes the details of the system at hand. The fields $\varphi_i^a$ are in general complex valued. The noise action (3) can be diagonalized by a linear transformation $\varphi_i^a = \sum_{bj} O_{ij}^{ab} \phi_j^b$ where $O_{ij}^{ab}$ is defined by $iJO^T \mathcal{J}^{-1}O/2 = \mathbb{1}$; here $O_{ij}^{ab}$ and $(\mathcal{J}^{-1})_{ij}^{ab}$ are treated as matrices by grouping the $(a, i)$ and $(b, j)$ indices. This yields [22–24]

$$S_0[\phi] = \frac{1}{2} \int_0^t dt' \sum_{ia} \phi_i^a(t') \phi_i^a(t') . \tag{5}$$

The real-valuedness of the fields $\phi_j^a$ is required for convergence of the integral (2) and can be viewed as defining the appropriate integration lines for the complex-valued fields $\varphi_j^a$ [24]. Due to the Gaussian action (5), the functional integral in Eq. (2) can be seen as an average over realizations of Gaussian white noise variables $\phi$ [22],

$$\hat{U}(t) = \langle \prod_j e^{\xi_j^+(t)\hat{S}_j^+} e^{\xi_j^z(t)\hat{S}_j^z} e^{\xi_j^-(t)\hat{S}_j^-} \rangle_\phi . \tag{6}$$

Eqs (4) can then be interpreted as stochastic differential equations (SDEs) [21–23]. Eq. (6) makes it possible to establish an exact map between quantum expectation values and classical averages. Time-evolved expectation values are given by $\mathcal{O}(t) = \langle \hat{U}^\dagger(t)\hat{O}\hat{U}(t)\rangle$, where $\langle \dots \rangle$ denotes the expectation value with respect to a chosen initial state. Quantum observables

can thus be cast in functional integral form by expressing their time dependence in terms of time-evolution operators $\hat{U}(t)$ and substituting the representation (6) [23, 24]. In order to disentangle two time-evolution operators, as in the case of local expectation values, one can introduce independent HS fields $\phi_x = \{\phi_{x,i}^a\}$ and the associated disentangling variables $\xi_x = \{\xi_{x,i}^a\}$, with $x \in \{f, b\}$ denoting forwards and backwards evolution. Eq. (6) features a product of single-site operators, whose action on product states can be straightforwardly evaluated; this yields a classical function $F_{\mathcal{O}}(\xi_i^a)$ such that $\langle \hat{\mathcal{O}} \rangle = \langle F_{\mathcal{O}}(\xi_i^a) \rangle_\phi$. Local observables correspond to classical functions of the form

$$F_{\mathcal{O}} = F_{\mathbb{1}} \bar{F}_{\mathcal{O}} \,, \tag{7}$$

where $\langle F_{\mathbb{1}}(t) \rangle_\phi$ gives the norm of the state and $\bar{F}_{\mathcal{O}}$ is a product of a finite number of terms, each featuring the disentangling variables relative to a single site [31]. For instance, for spin-1/2 systems the spin operators are represented by the Pauli matrices, $\hat{S}_i^a = \sigma_i^a/2$; inserting this in Eq. (6) one readily finds that for an initial $\otimes_i |\downarrow\rangle_i$ state the normalization function is given by

$$F_{\mathbb{1}}(t) \equiv \prod_i [1 + \xi_{f,i}^+(t)\xi_{b,i}^{+*}(t)] e^{-\frac{1}{2}[\xi_{f,i}^z(t) + \xi_{b,i}^{z*}(t)]} \,, \tag{8}$$

while the on-site longitudinal magnetization $\mathcal{M}_i^z(t) = \langle \psi(t) | \hat{S}_i^z | \psi(t) \rangle$ is given by $\mathcal{M}_i^z = \langle F_{\mathbb{1}} \bar{F}_{\mathcal{M}_i^z} \rangle_\phi$ with

$$\bar{F}_{\mathcal{M}_i^z}(t) = -\frac{1}{2} \left( \frac{1 - \xi_{f,i}^+(t)\xi_{b,i}^{+*}(t)}{1 + \xi_{f,i}^+(t)\xi_{b,i}^{+*}(t)} \right) \,. \tag{9}$$

Expressions such as (9) are easily obtained for any physical observable and take the same form for different models, in real or imaginary time; see Refs [24, 31]. Different initial conditions correspond to different $F_{\mathcal{O}}$ [23, 24], or, alternatively, can be encoded in the initial conditions of the disentangling variables [25, 31]. In general, any time-evolving quantity can be expressed within the disentanglement approach. However, the approach is best suited to quantities that are readily expressed in terms of time-evolution operators, since these are the objects that are replaced by their disentangled counterparts (2). This is the case of the local observables discussed above.

Quantum expectation values $\mathcal{O}(t)$ can be numerically computed by averaging the classical functions $F_{\mathcal{O}}$ over realization of the stochastic processes $\phi(t)$. As demonstrated in Refs [24, 25], the numerical evaluation of such averages is stymied by the exponential growth of fluctuations in $F_{\mathcal{O}}$ with time and the system size. In the imaginary-time case, it was recently shown that the growth of fluctuations can be greatly suppressed by applying an importance sampling method whereby, when randomly generating trajectories, strongly-contributing ones are sampled preferentially [31]. In contrast, when sampling according to the original measure (5) one mostly draws trajectories that are nearly non-interacting and give little contribution. Below we generalize the importance sampling approach to real-time evolution, showing that it leads to a significant reduction in fluctuations compared to the direct sampling approach of Refs [23–25].

## 3 Importance sampling

For analytical computations, it is convenient to work with the $\varphi$ fields. For a given observable $\mathcal{O}$, we seek to identify the saddle point (SP) trajectory $\varphi_{SP}$ yielding the largest contribution

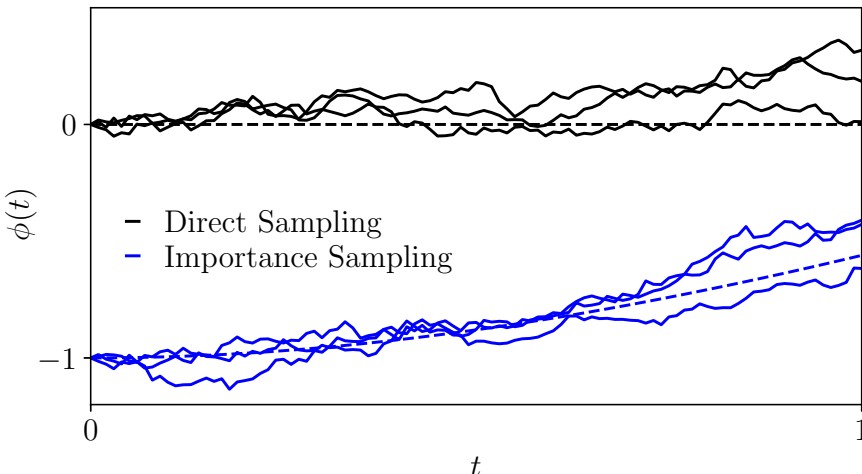

Figure 1: Illustration of the importance sampling approach. When directly sampling according to the measure (5), one predominantly generates trajectories around $\phi = 0$ (dashed black line), which might however carry a small contribution to a given observable. In contrast, in the importance sampling approach one preferentially generates trajectories in the vicinity of the saddle point trajectory (dashed blue line), which carries the largest contribution. The solid black and blue lines respectively represent stochastic trajectories generated according to direct and importance sampling.

to the corresponding functional integral. This is obtained by extremizing the effective action $S_{\mathcal{O}}[\varphi] \equiv S_0[\varphi] - \log F_{\mathcal{O}}[\varphi]$, defined such that [31]

$$\mathcal{O} = \int \mathcal{D}\varphi \, e^{-S_{\mathcal{O}}[\varphi]}. \tag{10}$$

The action $S_{\mathcal{O}}$ features the fields $\xi_i^a$ via the function $F_{\mathcal{O}}$; these are themselves functionals of $\varphi$, such that Euler-Lagrange equations cannot be derived. Instead, the SP equation is obtained by direct extremization of $S_{\mathcal{O}}$:

$$\left. \frac{\delta S_{\mathcal{O}}}{\delta \varphi(t)} \right|_{\varphi_{\text{SP}}} = 0. \tag{11}$$

This condition yields a functional integral equation for $\varphi_{\text{SP}}$, which can be solved recursively. One can then perform an exact measure transformation such that trajectories around the saddle point configuration are sampled preferentially, as illustrated in Fig. 1. This is carried out by performing the change of variables $\phi(t) \to \phi(t) + \phi_{\text{SP}}(t)$ in the functional integral (10) [31], where $\phi_{\text{SP}}$ is readily obtained from $\varphi_{\text{SP}}$ as indicated in Section 2. This transformation does *not* constitute a saddle point approximation: a change of measure does not truncate fluctuations, so that the resulting expressions are still formally exact. Due to the Gaussianity of the action (5), the importance sampling method then amounts to numerically sampling a modified functional

$$\langle \hat{\mathcal{O}} \rangle = e^{-S_0[\phi_{\text{SP}}]} \int \mathcal{D}\phi \, e^{-S_0[\phi]} e^{-\int dt \, \phi(t) \cdot \phi_{\text{SP}}(t)} F_{\mathcal{O}}[\phi_{\text{SP}} + \phi], \tag{12}$$

where $\phi \equiv \{\phi_{x,i}^a\}$, $\phi_{\text{SP}} \cdot \phi \equiv \sum_{iax} (\phi_{\text{SP}})_{x,i}^a \phi_{x,i}^a$, and the index $x$ runs over all sets of disentangling fields, e.g. forwards and backwards for local observables.

Eq. (12) can be evaluated numerically in order to compute quantum expectation values, yielding an importance sampling scheme for the stochastic approach. In practice, one generates an ensemble of stochastic trajectories $\xi_i^a$ whose time evolution is determined via the

SDEs (4) with the modified field $\phi(t) \to \phi(t) + \phi_{\mathrm{SP}}(t)$. To the best of our current knowledge, the SDEs (4) can only be solved analytically in certain special cases, such as non-interacting systems or Hamiltonians made up of commuting terms [24]. Thus, a discrete-time numerical method is generally needed in order to solve (4). Different numerical integration schemes have been previously used to this end, including the Euler-Maruyama [23,24,33] and the stochastic Heun [25,33,34] schemes. Here, unless otherwise stated, we use the explicit strong order-1 scheme [33,35] with time-step $\Delta t = 0.01$, which was found to perform comparatively well. The numerical time evolution of the disentangling variables $\xi_i^a$ is known to give rise to divergences whereby $|\xi_i^+(t)| \to \infty$ at finite $t$ [23]; this issue can be avoided by means of a suitable reparameterization of the disentangling variables [25,36]. We explicitly normalize observables by $\langle F_{\mathbb{1}} \rangle_\phi$, which improves the final accuracy [25]. For results obtained as classical averages of stochastic quantities, we estimate error bars as $\sigma/\sqrt{n_B}$, where $\sigma$ is the standard deviation over $n_B = 5$ batches of independent simulations unless otherwise specified.

For definiteness, we will illustrate the importance sampling approach by considering the quantum Ising model in $D$ spatial dimensions. For a system with $N = N_1 \times \cdots \times N_D$ sites, this is given by

$$\hat{H} = -J \sum_{\langle ij \rangle}^{N} \hat{S}_i^z \hat{S}_j^z - \Gamma \sum_{j=1}^{N} \hat{S}_j^x - h \sum_{j=1}^{N} \hat{S}_j^z, \tag{13}$$

where we use $D$-dimensional spatial indices $i = (i_1, \cdots, i_D)$ with $i_k \in \{1, \cdots, N_k\}$ and $\langle ij \rangle$ denotes pairs of nearest neighbors[1]. We consider periodic boundary conditions and ferromagnetic (FM) interactions, $J > 0$. When $D = 1$, the model (13) reduces to the quantum Ising chain, and for $h = 0$ it can be solved exactly in terms of free fermions, harboring a quantum phase transition (QPT) at $\Gamma_c = J/2$ in the present units [37]. The Hamiltonian (13) is encoded in the SDEs (4) with $h_j^+ = h_j^- = \Gamma/2$, $h_j^z = h$ [22,23]. In our numerical results below we set $J = 1$ and consider $D \in \{1, 2\}$, illustrating the applicability of the importance sampling method to higher-dimensional systems.

## 4 Local observables

In general, the SP equation (11) must be solved numerically, and the resulting SP field configuration depends on the chosen end time $t_f$, i.e. $\varphi_{\mathrm{SP}} \equiv \varphi_{\mathrm{SP}}(t|t_f)$. However, as further discussed below, for local observables of the form $\mathcal{O} = \langle \hat{U}^\dagger \hat{\mathcal{O}} \hat{U} \rangle$ the functional equation (11) can be reduced to a differential equation. The effective action for an observable of this form is given by

$$S_{\mathcal{O}} = S_0[\varphi_f] + S_0^*[\varphi_b] + \frac{1}{2} \sum_i \xi_{f,i}^z(t_f) + \frac{1}{2} \sum_i \xi_{b,i}^{z*}(t_f)$$
$$- \sum_i \log\left[1 + \xi_{f,i}^+(t_f)\xi_{b,i}^{+*}(t_f)\right] - \log \bar{F}_{\mathcal{O}}(t_f). \tag{14}$$

Eq. (14) features forwards and backwards fields $\varphi_x = \{\varphi_{x,j}^a\}$ with $x \in \{f, b\}$, introduced to decouple the two time-evolution operators in $\mathcal{O}$, and the respective noise actions $S_0[\varphi_x]$ and disentangling variables $\xi_{x,j}^a$, given by (3) and (4). Eq. (14) is general to spin-1/2 systems in any dimension. It is apparent that only the last term of Eq. (14) is observable-dependent.

---

[1]This model corresponds to $\mathcal{J}_{ij}^{ab} = \delta_{az}\delta_{bz} \sum_{d=1}^{D} \mathcal{J}_{ij}^d$, where $\mathcal{J}_{ij}^d = (\delta_{i_d j_d+1} + \delta_{i_d j_d-1})/2 \prod_{k \neq d} \delta_{i_k j_k}$ is the interaction matrix relative to the dimension $d$. For this model, if any of the dimensions $N_k$ of the system is a multiple of 4, the corresponding interaction matrix needs to be regularized by including a diagonal term, which does not affect the resulting dynamics; see Ref. [24].

Let us begin by considering the case $\bar{F}_{\mathcal{O}} = 1$, corresponding to the normalization function $F_{\mathbb{1}}$. Extremization of (14) with respect to $\varphi^a_{f,i}(t')$ leads to the SP equation

$$iJ \sum_{bk} [\mathcal{J}^{-1}]^{ab}_{jk} \varphi^b_{f,k}(t')\Big|_{\text{SP}} = -\frac{\delta \xi^z_{f,j}(t_f)}{\delta \varphi^a_{f,j}(t')}\Big|_{\text{SP}} + \frac{2\xi^{+*}_{b,j}(t_f)}{[1 + \xi^+_{f,j}(t_f)\xi^{+*}_{b,j}(t_f)]} \frac{\delta \xi^+_{f,j}(t_f)}{\delta \varphi^a_{f,j}(t')}\Big|_{\text{SP}}, \quad (15)$$

where we used $\delta \xi^a_j / \delta \varphi^b_k \propto \delta_{jk}$; explicit expressions for the functional derivatives are given in Appendix A. Symmetry between the forwards and backwards fields implies $\varphi^a_{f,i}|_{\text{SP}} = \varphi^a_{b,i}|_{\text{SP}} = \varphi^a_{\text{SP},i}$ and similarly $\xi^a_{f,i}|_{\text{SP}} = \xi^a_{b,i}|_{\text{SP}} \equiv \xi^a_{\text{SP},i}$. For ground state expectation values, SP equations analogous to (15) can be reduced to algebraic ones by considering the infinite imaginary-time limit [31]; this is however not possible in the present context of real-time evolution. However, it can be shown that the solution of Eq. (15) satisfies $\partial_{t_f} \varphi^a_{\text{SP},i}(t|t_f) = 0$; see Appendix B. As a consequence, the end time $t_f$ in Eq. (15) can be chosen freely so as to simplify the computation of the SP configuration. It is convenient to set $t_f = t'$; the SP equation (15) then readily yields

$$\varphi^a_{f,j}|_{\text{SP}} = \sum_k \mathcal{J}^{ab}_{jk} v^b_k, \quad (16)$$

where the vectors

$$v^a_k \equiv \frac{1}{1 + |\xi^+_{\text{SP},k}|^2} \left( \xi^{+*}_{\text{SP},k}, \frac{|\xi^+_{\text{SP},k}|^2 - 1}{2}, \xi^+_{\text{SP},k} \right), \quad (17)$$

with $a \in \{+, z, -\}$, feature the normalized expectation values of the spin operators $\hat{S}^a_k$ under the dynamics induced by the SP field. Eq. (16) is thus equivalent to a mean field condition, $\varphi^a_{\text{SP},j}(t) = \sum_{bk} \mathcal{J}^{ab}_{jk} \langle \hat{S}^b_k(t) \rangle|_{\text{SP}}$, whereby the effective field acting on each spin is produced by the magnetization of its neighbors. The above steps should in principle be repeated for each different observable, adding the corresponding term $-\log \bar{F}_{\mathcal{O}}$ to the action. However, for translationally invariant observables it can be shown that the $\bar{F}_{\mathcal{O}}$-dependent term in the SP equation becomes negligible in the thermodynamic limit; see Appendix C. This makes it possible to use the SP configuration given by (16) to perform importance sampling for local observables given a sufficiently large system. These findings generalize the results of Ref. [31], where it was shown that in the limit of infinite imaginary time the dominant contribution to ground-state expectation values corresponds to the mean-field ground state. The physical interpretation is analogous: the optimal approximation to the full dynamics of a quantum system within the manifold of single-spin trajectories, in the spirit of the time-dependent variational principle (TDVP) [38, 39], is given by mean field. In contrast to TDVP, however, here we do not restrict ourselves to the optimal trajectory, but perform a sum over trajectories: this restores entanglement, and the resulting time evolution is formally exact. The set of coupled equations (16) can be solved numerically together with (4); the solution matches the direct recursive solution of (15), but is much more efficient.

Having obtained the SP configuration from Eq. (16), we can perform importance sampling according to Eq. (12) to compute local observables. To illustrate the difference in performance between direct and importance sampling, in Fig. 2 we consider results obtained using the same numerical solution scheme, discretization time step, and number of simulations. We consider the longitudinal magnetization $\mathcal{M}^z = \sum_j \mathcal{M}^z_j$ of a $3 \times 3$ quantum Ising model initialized in the symmetry-broken ferromagnetic ground state for $\Gamma = 0$, $h < 0$, $|\Downarrow\rangle \equiv \otimes_i |\downarrow\rangle_i$, and evolved with $\Gamma = h = 2J$, comparing our results to exact diagonalization (ED) performed using the QuSpin package [40]. Generating each data set required $\approx 12$ minutes on a laptop, using 2 Intel Core i5 processors with a clock speed of 2.9 GHz. The results obtained using the importance sampling approach are in much better agreement with ED compared to direct sampling. The

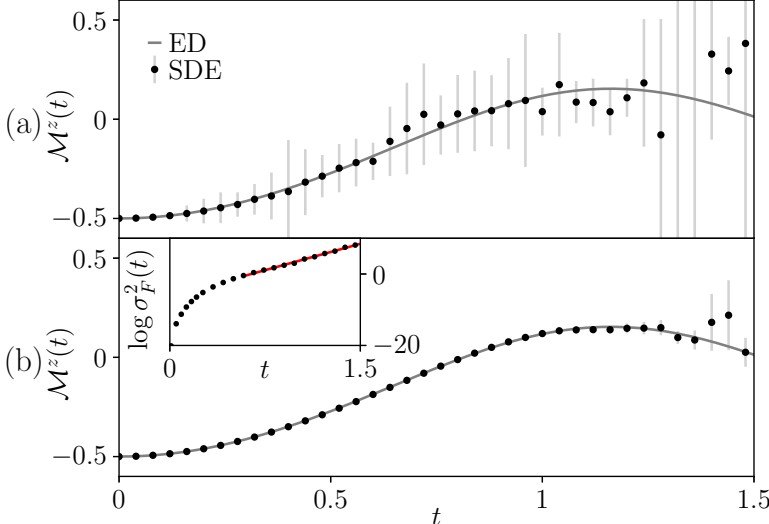

Figure 2: Time evolution of the longitudinal magnetization $\mathcal{M}^z$ of the 2D Ising model (13) following a quantum quench. We consider a $3 \times 3$ system initialized in the $|\Downarrow\rangle$ state and evolved using the Hamiltonian (13) with $\Gamma = h = 2J$. We compare the results obtained by solving the Ising SDEs (dots) using (a) direct sampling and (b) importance sampling, showing ED (full lines) as a benchmark. Each data set consists of $\mathcal{N} = 10^4$ trajectories. The error bars signal the much stronger and more rapidly growing fluctuations for direct sampling. The inset of panel (b) illustrates the exponential growth of fluctuations in the stochastic quantities with time; the solid red line shows the fit given in the main text.

difference between the two approaches is also reflected in the behavior of fluctuations around mean values. Due to the strong fluctuations in the direct sampling results, dividing the data set into small batches leads to an underestimation of fluctuations. Instead, in Fig. 2 we estimate fluctuations as the standard error $\sigma/\sqrt{\mathcal{N}}$ over the full data set, where the standard deviation $\sigma$ over $\mathcal{N}$ independent simulations is obtained from the standard deviations of the numerator and denominator by using uncertainty propagation [31]. The bars clearly show that the importance sampling scheme results in a significant mitigation of fluctuations. To investigate this quantitatively, in the inset of Fig. 2(b) we show the variance $\sigma_F^2$ of the stochastic function $F_{\mathbb{1}}$ yielding the normalization, whose behavior is representative of all local observables due to Eq. (7) [31]. Beyond a transient regime, the time evolution of this quantity is well-approximated by exponential growth, $\sigma_F^2(t) \sim \alpha \exp(\beta N t)$ with $\alpha \approx 10^{-3}$, $\beta \approx 1$. In contrast, for direct sampling one has a comparable $\beta \approx 1$ but a much larger prefactor $\alpha \approx 10$. Thus, the importance sampling approach does not eliminate the exponential growth of fluctuations, but it can suppress by several orders of magnitude the associated prefactor. This allows importance sampling to access larger systems and later times than it was previously possible using the direct approach [23–25]. As an example, in Fig. 3 we consider $5 \times 5$ and $13 \times 13$ systems initialized in the $|\Downarrow\rangle$ state and time-evolved using the Hamiltonian (13) with $\Gamma = h = J/4$. The suppression of fluctuations is increasingly effective as the transverse field $\Gamma$ is reduced and the classical $\Gamma = 0$ limit is approached, as previously reported for imaginary-time evolution [31]. This can again be understood in light of the physical interpretation of the importance sampling approach, whereby the sampling accounts for the presence of entanglement on top of the optimal mean-field trajectory. As a consequence, although the above derivation did not assume a particular regime, the approach can be expected to be most effective in regimes where mean

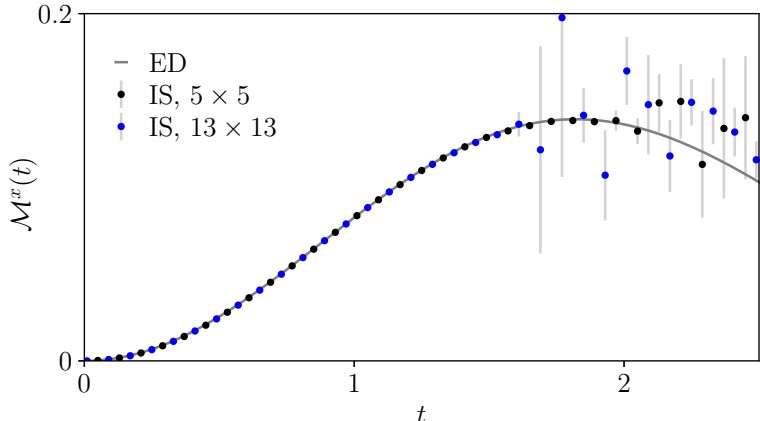

Figure 3: Time evolution of the transverse magnetization $\mathcal{M}^x$ in the 2D Ising model (13) initialized in the $|\Downarrow\rangle$ state and evolved using the Hamiltonian (13) with $\Gamma = h = J/4$. For a $5 \times 5$ system we compare the results obtained by importance sampling (IS) to ED, finding good agreement up to the time when fluctuations become sizable. We also include data for a $13 \times 13$ system, for which no ED results are available. The IS results were respectively obtained from $5 \times 10^4$ and $10^5$ independent trajectories. The error bars show the onset of large fluctuations at $t \approx 2.1$ and $t \approx 1.7$ for the chosen physical and computational parameters.

field theory would provide a reasonably good approximation to the true quantum dynamics.

# 5 Loschmidt amplitude

The importance sampling approach is not restricted to local observables and can be applied to compute global quantities. As an illustration, we consider the *Loschmidt amplitude* $A(t) \equiv \langle \psi(0)|\psi(t)\rangle$, where $|A(t)|^2$ gives the probability for the system to return to its initial state following unitary evolution for a time $t$. Since $A(t)$ is exponentially suppressed as $N \to \infty$, one typically considers the *rate function* $\lambda(t) \equiv -\log|A(t)|^2/N$, also known as the fidelity density, which has a well-defined thermodynamic limit [32]. This quantity has recently received great theoretical [8] and experimental [41] interest in the context of dynamical quantum phase transitions (DQPTs), a proposed generalization of equilibrium QPTs whereby $\lambda(t)$ becomes non-analytic as a function of time [8,32]. Notably, DQPTs can only occur in the thermodynamic limit, but signatures of their presence can be observed in large but finite systems [23, 24, 32, 41]. Furthermore, these phenomena typically occur at early times, making them promising candidates for currently available experimental platforms.

To illustrate how the importance sampling method can be applied to reveal DQPTs, we consider the 2D quantum Ising model (13) with $h = 0$. This model has a QPT at $\Gamma \approx 1.523\,J$ [42, 43]. We initialize the system in the symmetric FM ground state $|\psi_{\text{FM}}\rangle \equiv (|\Uparrow\rangle + |\Downarrow\rangle)/\sqrt{2}$, where $|\Uparrow\rangle \equiv \otimes_i |\uparrow\rangle_i$, and consider a quench deep into the paramagnetic phase; the return probability is then given by

$$|A|^2 = |\langle\Downarrow|\hat{U}|\Downarrow\rangle|^2 + |\langle\Uparrow|\hat{U}|\Downarrow\rangle|^2 \equiv |A_{ud}|^2 + |A_{dd}|^2, \tag{18}$$

where $\hat{U} = \hat{U}(t)$ and we used $|\langle\Uparrow|\hat{U}|\Uparrow\rangle| = \langle\Downarrow|\hat{U}|\Downarrow\rangle|$ and $|\langle\Uparrow|\hat{U}|\Downarrow\rangle| = |\langle\Downarrow|\hat{U}|\Uparrow\rangle|$. DQPTs in this quantity can be understood as arising from the crossing of the contributions of $|A_{ud}|^2$ and

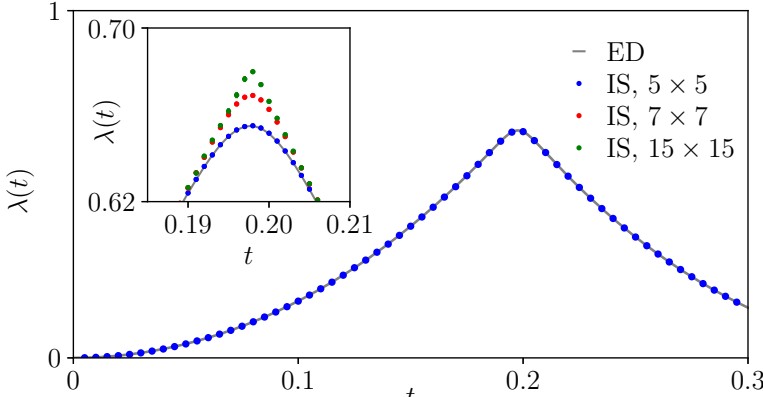

Figure 4: Time evolution of the Loschmidt rate function $\lambda(t)$ corresponding to the return probability (18) for a system initialized in the $\Gamma = h = 0$ ferromagnetic ground state $|\psi_{\text{FM}}\rangle$ and evolved using the Hamiltonian (13) with $\Gamma = 8J$, $h = 0$. The main panel shows that the results obtained from the importance sampling (IS) approach (dots) are in good agreement with ED (full line) for a $5 \times 5$ system. As the system size is increased, the peak in the rate function sharpens, eventually becoming non-analytic in the thermodynamic limit; this is demonstrated in the inset, where we show results for $7 \times 7$ and $15 \times 15$ systems for which ED cannot be performed. The numerical results were obtained from $5 \times 10^4$, $10^5$ and $10^6$ independent simulations, in order of increasing system size. The error bars are nearly invisible on the scale of the plot.

$|A_{dd}|^2$. In the stochastic approach, the amplitudes are obtained as $A = \langle f \rangle_\phi$ with

$$f_{dd}(t) = e^{-1/2 \sum_i \xi_i^z(t)/2}, \quad f_{ud}(t) = f_{dd}(t) \prod_i \xi_i^+(t). \tag{19}$$

The SP equation for each amplitude can be obtained as in (11) and solved numerically to find the SP field. In contrast to local observables, here the SP field is *not* equal to the mean field and the whole configuration depends on the chosen end time $t_f$, $\varphi_{\text{SP}} \equiv \varphi_{\text{SP}}(t|t_f)$.

One can then perform importance sampling separately for the two contributions in (18), using the SP field obtained numerically for each. Fig. 4 shows the results obtained in this way for systems of increasing size. In the main panel we compare our results to ED for a $5 \times 5$ system time-evolved with $\Gamma = 8J$, finding good agreement. Here we use the stochastic Heun scheme applied in Ref. [25], setting $\Delta t = 10^{-4}$. Notably, the number of simulations required by the importance sampling scheme to accurately reproduce the ED result is more than two orders of magnitudes smaller in comparison to direct sampling using the measure (5) [25]. The inset shows that the peak becomes increasingly sharp as the system size is increased to $7 \times 7$ and $15 \times 15$.

It is known that non-analytic points can also occur in the time evolution of the term $|A_{dd}|^2$, which corresponds to the return probability for a quench from the symmetry-broken initial state $|\Downarrow\rangle$, and the relative rate function $\lambda_{dd} = -\lim_{N \to \infty} \log |A_{dd}|^2/N$ [32]. In contrast to the previously considered case of a quench from $|\psi_{\text{FM}}\rangle$, such DQPTs cannot be straightforwardly understood as arising from a sum of competing contributions. However, insights about the origin of DQPTs in the amplitude $|A_{dd}|$ can be obtained from the solution of the SP equation itself. We illustrate this for the 1D Ising chain, where the exact location of the non-analytic point can be computed analytically or determined to arbitrary numerical precision using infinite time-evolving block decimation (iTEBD) [12].

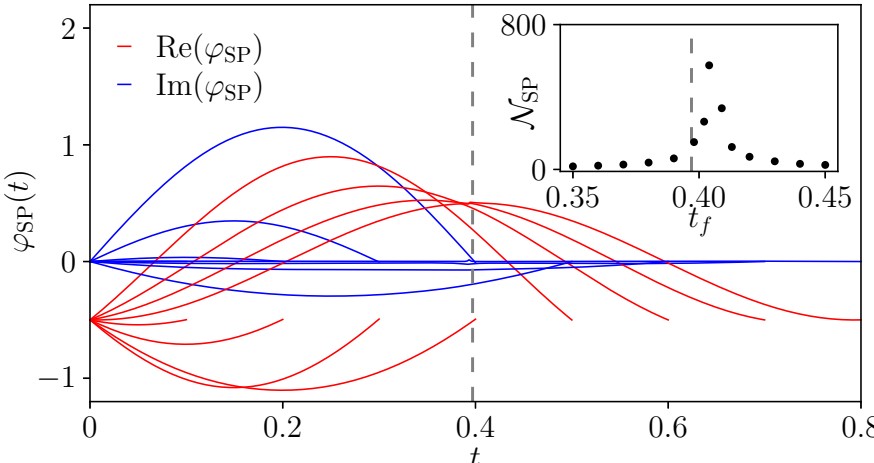

Figure 5: Time evolution of the SP field $\varphi_{SP}(t)$ corresponding to the rate function $\lambda_{dd}(t_f)$ for a quantum Ising chain initialized in the $|\Downarrow\rangle$ state and evolved with $\Gamma = 8J$, $h = 0$. The red and blue lines respectively show the real and imaginary parts of $\varphi_{SP}(t)$ for different end times $t_f$, where $t_f$ corresponds to the point at which each curve terminates. In contrast to the case of local observables, the SP field shows a marked dependence on the end time. Notably, the SP field changes abruptly when the end time is chosen to be near the location of a DQPT at $t^* \approx 0.397$ (vertical dashed line), with its real and imaginary parts changing sign. The inset shows that the number $\mathcal{N}_{SP}$ of iterations required to solve the SP equation sharply increases in a region close to the DQPT.

In Fig. 5 we consider the solution $\varphi_{SP}(t|t_f)$ of Eq. (11) as a function of the stopping time $t_f$ for a system initialized as $|\Downarrow\rangle$ and evolved with $\Gamma = 8J$. For this quench, a DQPT occurs at $t^* \approx 0.397$. It can be seen that the entire SP configuration evolves as a function of $t_f$, with an abrupt change occurring at $t_f \approx t^*$ whereby the real and imaginary parts of the SP field change sign. As shown in the inset, the number of iterations $\mathcal{N}_{SP}$ required to recursively solve Eq. (11) sharply increases in a region near $t_f \approx t^*$. In particular, we find that the recursive solution does not converge within $10^3$ iterations for $0.404 < t_f < 0.409$. Such behavior points to an instability of the recursive solution in this region, which could be due to the coexistence of different minima yielding a comparable action; the leading and subleading contributions would then switch at a critical time $t_{SP}^* \approx 0.406$. These observations suggest that DQPTs in $|A_{dd}|^2$ are associated with an abrupt change of the product-state configuration carrying the largest contribution to the quantum dynamics, consistently with recent findings [44]. This immediately generalizes the phenomenology of ground-state QPTs in the disentanglement formalism, where QPTs are associated with an abrupt change of the dominant trajectory as a function of a control parameter [31]. Similar switching behavior near DQPTs was previously observed by considering generalized expectation values in fermionic systems [45]. In the present context, this phenomenology would not be reproduced if the SP field corresponded to the mean field, since it is closely tied to the end-time dependence of Eq. (11). The full quantum dynamics, obtained from sampling, includes the additional effects of entanglement on top of the optimal SP product-state dynamics. This potentially explains the small difference between $t_{SP}^*$ and the DQPT critical time $t^*$: the product-state approximation provided by the SP field can only determine the DQPT location approximately, as a degree of entanglement is always needed to fully capture these phenomena [44].

# 6 Conclusions

In this manuscript, we have introduced an importance sampling scheme for the real-time dynamics of many-body quantum spin systems. The importance sampling method is based on the disentanglement approach, whereby unitary quantum dynamics is exactly mapped to an ensemble of classical stochastic processes. Quantum expectation values are then obtained as averages over stochastic trajectories, which can be generated numerically. We have shown that the dominant contribution to a given observable is given by a *saddle point trajectory*, which can be obtained by extremizing an appropriate effective action. By preferentially sampling trajectories close to the saddle point trajectory, it is possible to significantly improve the performance of the method compared to direct sampling, as we have demonstrated for both local observables and return probabilities. This improvement in performance is due to a suppression in the strength of fluctuations in the stochastic quantities, which determines the numerical efficiency of the method. However, fluctuations are still found to grow exponentially with time and the system size; while the accessible parameter regions can be extended by using importance sampling, large systems and late times remain challenging to capture. Further progress will thus be needed in order for the method to access interesting regimes of higher-dimensional quantum dynamics; several directions for development can be envisaged, including cluster approaches and the development of approximate schemes that truncate fluctuations. Since the importance sampling scheme completely eliminates fluctuations in the limit of a Hamiltonian made up of commuting terms, it would also be interesting to explore its application to higher-spin systems, where the performance of the approach might benefit from the proximity to a classical limit. Furthermore, the broader framework introduced in this work might prove useful beyond its application to importance sampling. In the context of imaginary-time evolution, it has recently been shown that corrections to the mean-field estimate for ground state expectation values can be analytically obtained order-by-order by viewing the disentanglement approach as a field theory [31]. A similar development for real-time evolution would make it possible to systematically include the effect of entanglement on top of the leading-order mean-field dynamics.

*Acknowledgments.*— SDN would like to thank S. Begg, M. J. Bhaseen, B. Doyon, V. Gritsev, C. Mendl and M. Serbyn for valuable feedback and discussions. SDN acknowledges funding from the Institute of Science and Technology (IST) Austria, and from the European Union's Horizon 2020 research and innovation programme under the Marie Skłodowska-Curie Grant Agreement No. 754411.

During the preparation of this manuscript we became aware of the work [46], in which an importance sampling scheme is developed by considering the Hermiticity of the effective Hamiltonian governing the stochastic evolution; this also leads to an improvement in the accessible time-scale for a given number of simulations.

# A Functional derivatives of the disentangling variables

Here we provide the functional derivatives of the disentangling variables, which can be obtained from the SDEs (4). For clarity we suppress site indices, $\xi_i^a \to \xi^a$, since all variables at different sites are independent. Let us begin from $\xi^+$; differentiating the equation of mo-

tion (4a) yields

$$\frac{\delta \dot{\xi}^+(t)}{\delta \varphi^a(t')} = iJ\delta(t-t')\big[\delta_{a+} + \xi^+(t)\delta_{az} - [\xi^+(t)]^2\delta_{a-}\big] + i\big[\Phi^z(t) - 2\Phi^-(t)\xi^+(t)\big]\frac{\delta \xi^+(t)}{\delta \varphi^a(t')},$$
(20)

resulting in

$$\frac{\delta \xi^+(t)}{\delta \varphi^a(t')} = iJ\theta(t-t')\big[\delta_{a+} + \xi^+(t')\delta_{az} - [\xi^+(t')]^2\delta_{a-}\big]\exp\left(i\int_{t'}^t \big[\Phi^z(s) - 2\Phi^-(s)\xi^+(s)\big]ds\right).$$
(21)

Proceeding similarly for $\xi^z$, Eq. (4b) yields

$$\frac{\delta \dot{\xi}^z(t)}{\delta \varphi^a(t')} = iJ\delta(t-t')\big[\delta_{az} - 2\xi^+(t)\delta_{a-}\big] - 2i\Phi^-(t)\frac{\delta \xi^+(t)}{\delta \varphi^a(t')},$$
(22)

which integrates to

$$\frac{\delta \xi^z(t)}{\delta \varphi^a(t')} = iJ\theta(t-t')\big[\delta_{az} - 2\xi^+(t')\delta_{a-}\big] - 2i\int_{t'}^t \Phi^-(s)\frac{\delta \xi^+(s)}{\delta \varphi^a(t')}ds.$$
(23)

## B  Saddle point field for the normalization

The SP equation (15) was obtained by extremizing the action (14) for $\mathcal{O} = \mathbb{1}$ with respect to the field $\varphi_{f,i}^a$. Saddle point configurations can in principle depend on the chosen end time $t_f$, $\varphi_{\mathrm{SP},i} \equiv \varphi_{\mathrm{SP},i}(t|t_f)$, as is the case in Section 5. However, a recursive numerical solution of Eq. (15) shows that its solution is independent of $t_f$, which provides a significant simplification. In order to show this analytically, we differentiate Eq. (15) with respect to $t_f$, using the explicit expressions (21), (23). Let us collect into the left-hand side $L$ all terms featuring explicit derivatives with respect to $t_f$ of variables evaluated at times $t' \neq t_f$, namely all terms proportional to $\partial_{t_f}\varphi_{\mathrm{SP},i}^a(t')$, $\partial_{t_f}\xi_{\mathrm{SP},i}^a(t')$ with $t' < t_f$. This term reads:

$$L/J = \sum_{bk}[\mathcal{J}^{-1}]_{jk}^{ab}\partial_{t_f}\varphi_{\mathrm{SP},k}^b(t') - 2\delta_{a-}\partial_{t_f}\xi_{\mathrm{SP},j}^+(t')$$
$$+ \frac{2\xi_{\mathrm{SP},j}^{+*}(t_f)}{[1 + \xi_{\mathrm{SP},j}^+(t_f)\xi_{\mathrm{SP},j}^{+*}(t_f)]}\exp\left(i\int_{t'}^{t_f}\big[\Phi_{\mathrm{SP},j}^z(s) - 2\Phi_{\mathrm{SP},j}^-(s)\xi_{\mathrm{SP},j}^+(s)\big]ds\right)$$
(24)
$$\times\left[\left(\delta_{az} - 2\delta_{a-}\xi_{\mathrm{SP},j}^+(t')\right)\partial_{t_f}\xi_{\mathrm{SP},j}^+(t') + i\int_{t'}^{t_f}\partial_{t_f}[\Phi_{\mathrm{SP},j}^z(s) - 2\xi_{\mathrm{SP},j}^+(s)\Phi_{\mathrm{SP},j}^-(s)]ds\right].$$

The remaining terms are collected into the right-hand side $R$:

$$R = 2\Phi_{\mathrm{SP},j}^-(t_f)\frac{\delta \xi_{f,j}^+(t_f)}{\delta \varphi_{f,j}^a(t')}\bigg|_{\mathrm{SP}} + 2\frac{\delta \xi_{f,j}^+(t_f)}{\delta \varphi_{f,j}^a(t')}\bigg|_{\mathrm{SP}}$$
$$\times\left[\frac{i\xi_{\mathrm{SP},j}^{+*}(t_f)\big[\Phi_{\mathrm{SP},j}^z(t_f) - 2\Phi_{\mathrm{SP},j}^-(t_f)\xi_{\mathrm{SP},j}^+(t_f)\big]}{1 + \xi_{\mathrm{SP},j}^+(t_f)\xi_{\mathrm{SP},j}^{+*}(t_f)} + \frac{\partial}{\partial t_f}\frac{\xi_{\mathrm{SP},j}^{+*}(t_f)}{[1 + \xi_{\mathrm{SP},j}^+(t_f)\xi_{\mathrm{SP},j}^{+*}(t_f)]}\right].$$
(25)

Using the equations of motion (4), the right-hand side simplifies to

$$R = 2 \frac{\delta \xi_{f,j}^+(t_f)}{\delta \varphi_{f,j}^a(t')}\Big|_{\text{SP}} \Bigg[ \Phi_{\text{SP},j}^-(t_f) + \tag{26}$$

$$\frac{\Phi_{\text{SP},j}^{-*}(\xi_{\text{SP},j}^{+*})2 - \Phi_{\text{SP},j}^+(\xi_{\text{SP},j}^{+*})^2 - \Phi_{\text{SP},j}^{+*} - \Phi_{\text{SP},j}^- \left[ (\xi_{\text{SP},j}^+)^2(\xi_{\text{SP},j}^{+*})^2 + 2\xi_{\text{SP},j}^+\xi_{\text{SP},j}^* \right] + \Phi_{\text{SP},j}^z \xi_{\text{SP},j}^{+*} - \Phi_{\text{SP},j}^{z*}\xi_{\text{SP},j}^{+*}}{[1 + \xi_{\text{SP},j}^+\xi_{\text{SP},j}^{+*}]^2} \Bigg].$$

It is easy to see that the right-hand side $R$ vanishes provided $\Phi_{\text{SP},i}^+ = (\Phi_{\text{SP},i}^-)^*$ and $(\Phi_{\text{SP},i}^z)^* = \Phi_{\text{SP},i}^z$, which, for a Hermitian Hamiltonian (1) such that $h_i^+ = (h_i^-)^*$, corresponds to the conditions $\varphi_{\text{SP},k}^+ = (\varphi_{\text{SP},k}^-)^*$, $\varphi_{\text{SP},k}^z \in \mathbb{R}$; the solution for $\varphi_{\text{SP},k}^a$ obtained below can be self-consistently checked to satisfy these conditions. The initial equality is then verified if $\partial_{t_f} \varphi_{\text{SP},i}^a(t') = 0 \, \forall \, t' \neq t_f$, so that $L$ also vanishes; a $t_f$-independent solution is thus consistent with Eq. (15). We therefore arbitrarily choose $t_f$ in (15); the simplest choice is given by $t_f = t'$, which yields:

$$\sum_{bk} [\mathcal{J}^{-1}]_{jk}^{ab} \varphi_{\text{SP},k}^b = -\frac{1}{2}\left[ \delta_{az} - 2\delta_{a-}\xi_{\text{SP},j}^+ \right] + \frac{\xi_{\text{SP},j}^{+*}}{1 + \xi_{\text{SP},j}^+\xi_{\text{SP},j}^{+*}}\left[ \delta_{a+} + \delta_{az}\xi_{\text{SP},j}^+ - \delta_{a-}(\xi_{\text{SP},j}^+)^2 \right], \tag{27}$$

where the explicit time-dependence has been suppressed, since all variables are evaluated at the same time $t$, and we used Eqs (21) and (23). Considering the different cases $a \in \{+, z, -\}$ readily reproduces Eq. (16), which is consistent with the conditions $\varphi_{\text{SP},k}^+ = (\varphi_{\text{SP},k}^-)^*$, $\varphi_{\text{SP},k}^z \in \mathbb{R}$. This can be checked to be a solution of (15) by substitution. For the quantum Ising model, a direct numerical solution of Eq. (15) matches the solution (16).

## C  Saddle point field for translationally invariant observables

In the main text we consider the SP equation corresponding to the normalization function $F_{\mathbb{1}}$. Here we show that, in the thermodynamic limit, the same SP trajectory applies to any other translationally invariant observable. In general, a local observable is given by

$$\langle \hat{\mathcal{O}} \rangle = \langle F_{\mathbb{1}} F_{\mathcal{O}} \rangle_\phi, \tag{28}$$

where $F_{\mathcal{O}}$ is a classical function determined by the observable $\hat{\mathcal{O}}$. Crucially, $F_{\mathcal{O}}$ for translationally invariant local observables is a sum of $N$ terms, each featuring the variables $\xi_i^a$ at a single site. For instance, for the $z$-magnetization $\mathcal{M}^z = \langle \sum_i \hat{S}_i^z \rangle / N$ one has

$$F_{\mathcal{M}^z} = -\frac{1}{2N} \sum_j \frac{1 - \xi_{f,j}^+\xi_{b,j}^{+*}}{1 + \xi_{f,j}^+\xi_{b,j}^{+*}}. \tag{29}$$

The effective action for the observable $\mathcal{O}$ is given by

$$\mathcal{S}_{\mathcal{O}} = \mathcal{S}_{\mathbb{1}} - \log F_{\mathcal{O}}. \tag{30}$$

Proceeding as for the normalization leads to the saddle point equation for the field $\varphi_{\text{SP},j}^{(\mathcal{O})}(t')$

$$iJ \sum_{kb} [\mathcal{J}^{-1}]_{jk}^{ab} \varphi_{f,k}^{(\mathcal{O})b}\Big|_{\text{SP}} = (\mathcal{F}^{\mathbb{1}})_{f,j}^a - \frac{2}{f_{\mathcal{O}}} \frac{\delta F_{\mathcal{O}}}{\delta \varphi_{f,j}^{(\mathcal{O})a}}\Big|_{\text{SP}}, \tag{31}$$

where $\mathcal{F}^{\mathbb{1}}$ is is the right-hand side of Eq. (15), which gives the SP condition for the normalization. We can thus write

$$iJ\varphi_{f,j}^{(\mathcal{O})a}\Big|_{\text{SP}} = \sum_{kb}\mathcal{J}_{jk}^{ab}(\mathcal{F}^{\mathbb{1}})_{f,k}^{b} - \frac{1}{F_{\mathcal{O}}}\sum_{kb}\mathcal{J}_{jk}^{ab}\frac{\delta F_{\mathcal{O}}}{\delta\varphi_{f,k}^{(\mathcal{O})b}}\Big|_{\text{SP}}. \tag{32}$$

Consider the second term on the right-hand side: at the saddle point, $F_{\mathcal{O}}$ is of order 1, while for local models the sum only runs over few terms due to the sparse structure of $\mathcal{J}_{jk}^{ab}$; each of these terms is of order $1/N$. Therefore, this term does not contribute in the thermodynamic limit, and, for sufficiently large systems, one can use the SP field obtained by solving Eq. (15) to perform importance sampling for local observables.

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
