# Peer review of "Importance Sampling Scheme for the Stochastic Simulation of Quantum Spin Dynamics"

_SciPost Physics, doi:SciPost Phys. 11, 048 (2021)_

## Round 1 · Referee Report · Anonymous (Referee 1) · 2021-5-12

Report

This work deals with an algorithmic development for the stochastic simulation of the real-time dynamics of quantum spin systems. In recent years a numerical approach has been introduced (involving also the present author) where the dynamics generated by Schrödinger's equation is mapped onto a classical stochastic process. While this mapping is asymptotically exact upon taking into account sufficiently many classical noise trajectories, it has turned out that the initially introduced noise sampling scheme is facing a severe limitation in that the realization-to-realization fluctuations have appeared to grow exponentially both with system size and simulation time. In the end this has naturally led to strong constraints on this numerical approach.

In the present work the author introduces an importance sampling scheme which significantly reduces the resulting fluctuations by orders of magnitude. These advances are certainly important for the approach and significantly extend its regime of applicability. This is also demonstrated in the present manuscript where the short-time dynamics of Ising models on 2D lattices up to a size of 15x15=225 spins has been computed, which I consider a convincing achievement. As the author shows, the fluctuations still grow exponentially in size and time. Importantly, however, the importance sampling scheme allows to reduce the prefactor of the growth by several orders of magnitude. Although the approach therefore still faces its limitation due to an exponential growth of the required computational resources, this is in line with many other state-of-the-art numerical approaches to solve Schrödinger's equation, especially for 2D quantum matter. Consequently, even being able to access transient dynamics in this regime is an achievement, which certainly warrants publication in SciPost in my opinion.

The paper is nicely written. It provides a very good introduction to the field and the method. First the method is applied to a 2D Ising model by computing the magnetization dynamics and comparing the different sampling schemes. In a second step the author also considers the Loschmidt amplitude and identifies a dynamical quantum phase transition, signaled by temporal singular behavior in that quantity. In this context the emergence of this temporal critical behavior is set into context with an abrupt switching of the dominant contributing trajectories. This highlights another strength of the manuscript which at various instances tries to interpret the numerical solutions in terms of physical consequences.

In view of the current efforts towards accessing the real-time dynamics of 2D quantum matter both theoretically and experimentally, I consider the contributions of the present manuscript important. As a consequence I am convinced that the work deserves publishing in SciPost and I recommend publication in its present form.

I have just one final question for my own curiosity: is it possible to interpret the exponential growth of fluctuations as a kind of Lyapunov exponent of the underlying nonlinear classical differential equations?

---

## Round 1 · Referee Report · Anonymous (Referee 2) · 2021-5-24

Strengths

  1. New approach to studying dynamics of quantum spin systems.
  2. Relevant to both theorists and experimentalists.
  3. Approach seems to be powerful and versatile.
  4. Paper nicely written

Weaknesses

  1. Lack of adequate benchmarking against known results.
  2. Incompletediscussion on the applicability of the method - when is it applicable, what observables are measurable, etc.

Report

The author has developed an importance sampling method to study real-time dynamics of quantum spin systems based on a disentanglement approach. This is a generalization of a similar scheme introduced earlier for imaginary time dynamics where unitary quantum dynamics is mapped to an ensemble of classical stochastic processes. The author shows that an importance sampling of trajectories close to the saddle point trajectory can significantly reduce the fluctuations in the measured expectation values of physical observables, compared to direct sampling approach.

This is an interesting work. The study of dynamics in quantum many body systems is a challenging problem. At the same time it is crucial in gaining deeper insight into quantum phases and phenomena as well as understanding several experiments. In this context, the current work is timely, relevant and will be of interest to many researchers in the field. There is no reason to doubt the correctness of the results. However, this referee feels that the manuscript can be significantly improved if the author addresses the following:

  1. Can the author calculate the dynamic structure factor (DSF) using his newly developed approach? This will provide an important benchmark for the new method as the DSF for several key models are known from alternative approaches. It will also open up the new method to model inelastic neutron scattering experiments.

  2. Can the author discuss whether the new method can be used to calculate observables such as the Berry curvature or fidelity?

  3. The author states that the divergence in the Loschmidt ratio (LR) is related to the presence of a QCP in the ground state of the Ising model in a transverse field. Is it possible to establish a formal correspondence for the divergence of LR in any generic model and a corresponding QCP in the corresponding ground state phase diagram? Can critical exponents be extracted from the LR?

  4. Is the current approach limited to weak interactions where a mean field / classical solution can be written down in terms of a single particle basis? Or is it applicable generically? A short comment on the range of applicability will be useful.

In addition to the above there are a few minor comments:

  1. How are \phi(t) and \varphi(t) related?
  2. Has the author studied the time evolution of a coherent state?
  3. As the transverse field is reduced, the Hamiltonian approaches an exactly solvable limit. Is that connected to the accompanying suppression of fluctuations?
  4. Why does the author choose to use only odd system sizes? Does he use periodic or open boundary conditions?
  5. It would be better to mention the values of the Hamiltonian parameters used when discussing the results presented in a particular figure (right now, the complete set of parameters are given in the figure captions.
  6. It would be more consistent to use either imaginary / real time or Euclidean / Lorentzian time – the former being possibly more readily understood by the broader community.

The referee feels that the manuscript contains important results that deserve to be published. Addressing the above comments will make it more impactful.

Requested changes

Please refer to the previous section.

---

## Round 2 · Referee Report · Anonymous (Referee 2) · 2021-7-3

Report

The author has addressed all my queries satisfactorily. I believe the manuscript is suitable for publication in its current form.

---

## Round 2 · Referee Report · Anonymous (Referee 1) · 2021-7-14

Report

The author has satisfactorily addressed my last remaining question, which has anyway been more related to my own scientific curiosity than targeting the quality of the manuscript. Along the lines of my previous report I very much recommend publication of the paper.

---

## Round 2 · Author Response

Dear Editor and Referees,

I would like to thank you for your careful analysis and positive assessment of the present manuscript and for the Referees’ helpful and supportive comments. Answers to the Referees’ questions and comments are provided below and in the revised manuscript, as indicated in the list of changes.

Reply to Referee 1:

I would like to thank the Referee for their attentive reading of the manuscript and their very positive and supportive comments. Below I answer the Referee’s question.

  1. The Referee asks whether the exponential growth of fluctuations in the stochastic quantities can be interpreted in terms of a Lyapunov exponent (LE). To the best of my knowledge, LEs are typically defined for differential equations in order to quantify the divergence of solutions upon a small perturbation of the initial state, i.e. for trajectories ( x ) one has a LE ( \lambda ) when ( |\delta x(t) | \approx e^{\lambda t} |\delta x_0| ) . A positive maximal LE then signals (deterministic) chaotic behavior. In the present context, in contrast, all trajectories are initialized with identical initial conditions for all realizations of the SDEs, ( \delta \xi^a_i(0) =0 ), and the trajectories diverge only as a consequence of the subsequent stochastic evolution. Therefore, in the present case a LE in the above sense cannot be defined; however, it might be possible to quantify the degree of divergence in the stochastic trajectories by a suitable generalization of this quantity. I would like to thank the Referee for suggesting this connection, which could be an interesting direction for future work.

Reply to Referee 2:

I would like to thank the Referee for their attentive reading of the manuscript and for providing valuable inputs for clarification and further improvement. Below I answer the Referee’s questions and comments.

1 & 2. The dynamic structure factor can in principle be computed similarly to how it is done from other numerical techniques, e.g. tensor network methods, by first computing the required 2-point functions and then numerically performing the required time integral and spatial sum, see e.g.: S. R. White and I. Affleck, Phys. Rev. B 77, 134437 (2008). Nonetheless, in practice only short times, corresponding to high frequencies, can presently be accessed; thus, an extrapolation method would currently be needed, the development of which is beyond the scope of the present work. However, it is worth noticing that this shortcoming is shared with the majority of numerical methods, including tensor-network based approaches, which are similarly limited to short times for 2D systems. In spite of this limitation, several benchmarking possibilities are however still available for the approach. These include comparisons to exact diagonalization, as presently done in the manuscript, which gives access to arbitrarily late times for systems of up to 20-25 spins, and comparisons to other numerical methods that are currently being developed, such as those of Refs [17-20]. Recent developments in the field of ultracold atoms and trapped ions (see e.g. Refs [1-2]) could also make it possible to benchmark the importance sampling approach against experimental results by directly comparing quantities such as local expectation values. The Berry connection can be computed by performing adiabatic time evolution and numerically differentiating the instantaneous eigenstates with respect to the evolution’s driving parameters; the Berry curvature then requires an additional differentiation. This is again possible in principle within the present approach. However, the computational implementation of this is beyond the scope of the present manuscript, which is concerned with the development of the importance sampling method and a number of immediate benchmarks. Nonetheless, I would like to thank the Referee for bringing attention to these important quantities, which in future works could provide additional benchmarks and open up new applications of the disentanglement approach. In contrast, the fidelity (i.e. return probability) is one of the simplest quantities to compute within the stochastic approach, and it corresponds to the quantity ( |A(t)|^2 ) considered in Section V. In general, although any time-dependent quantity can be represented in this language, the stochastic approach is most suited to computing quantities that can be readily expressed in terms of the time-evolution operator, such as local expectation values. A comment clarifying this was added after Eq. (9).

  1. For the transverse-field Ising model, it can be shown analytically that DQPTs occur upon quenching across a QCP, as demonstrated in Ref. [32]. However, this correspondence is not general: examples are known of DQPTs occurring when quenching within a phase, as well as of the absence of DQPTs in spite of quenching across a QCP, see e.g.: S. Vajna and B. Dóra, Phys. Rev. B 89, 161105(R) (2014). Similarly, critical exponents relative to DQPTs can be defined, see Ref. [8], but they do not appear to have a general relation to equilibrium exponents.

  2. The current approach does not make assumptions about the strength of interactions; it is generally applicable to Hamiltonians of the form (1). Based on the derivation given in the manuscript, the optimal trajectory will be given by a mean-field trajectory in the single-spin basis. This is ultimately due to the fact that the disentanglement approach represents the system as an ensemble of non-interacting spins under the action of external fields. So, although the importance sampling approach is more generally applicable, it can be expected to be most advantageous in cases where mean field provides a reasonably good approximation to the true quantum dynamics. This is because the sampling has to account for the part of the dynamics that differs from mean field (i.e., for entanglement on top of the mean field evolution). This is now further clarified at the end of Section IV. This also lies at the root of the better performance observed for weak fields, since this regime approaches a classical limit, as remarked by the Referee in their minor comment 4; a comment clarifying this point was added at the end of Section IV. It is however possible that, for systems where a mean field approach would not work well, a similar importance sampling scheme could be formulated using a basis other than the single spin one. This would be an interesting direction for future developments, and I would like to thank the Referee for highlighting this point.

Reply to the minor comments:

  1. The relation between ( \varphi ) and ( \phi ) is given in the paragraph following Eq. (4). Namely, they are related by a linear transformation given by the matrix ( O ), which is determined by the interaction matrix ( \mathcal{J} ). Clarifications on this were added in the paragraphs following Eqs (4) and (11).

  2. In the present manuscript, for simplicity I focused on the time-evolution of the ( \downarrow ) spin coherent state. However, evolving any other spin coherent state poses no additional difficulty and can be done within the same formalism by simply adjusting the initial conditions ( \xi^a_i(0) ). This was previously shown in e.g. Ref. [25] or Ref. [31] for imaginary time evolution.

  3. This point was addressed within item 4 of the main comments.

  4. The stochastic approach can treat odd and even system sizes on an equal footing, see e.g. Ref. [23] where an Ising chain of 14 spins is considered. The importance sampling scheme does not add any additional complication relative to this. However, for the specific case of the Ising model additional care is needed as the interaction matrix is not invertible if any of the dimensions of the system is a multiple of 4. This is discussed in Ref. [24], where it is shown that this issue can be readily circumvented by introducing a diagonal shift to the interaction matrix ( \mathcal{J} ), which does not affect the ensuing dynamics. In the updated manuscript, a footnote was dedicated to a more detailed discussion of the explicit form of the interaction matrix ( \mathcal{J} ) for the Ising model, including a comment regarding sizes multiple of 4. A typo was also corrected. In the general derivations of the manuscript, no boundary conditions are assumed and the interaction matrix ( \mathcal{J} ) is allowed to be general. A comment clarifying this was added following Eq. (1). However, for numerical calculations we consider the Ising model (13) with periodic boundary conditions, as mentioned following Eq. (13).

5 & 6. I thank the Referee for suggesting these improvements; in the revised manuscript the missing physical parameters are now also provided in the main text of Sections IV and V, and the nomenclature has been uniformly changed to real/imaginary time.

---

## Round 2 · List of Changes

- Section I, after Eq. (1): added an explicit mention that no boundary conditions are assumed.
- Section II, after Eq. (4): added a comment on the definition of the \( O \) matrix.
- Section II, after Eq. (9): added a remark on which quantities can be expressed in the disentanglement formalism.
- Section III, after Eq. (11): added a comment pointing to the relevant discussion of the relation between \( \phi \) and \( \varphi \) fields.
- Section III, after Eq. (13): explicitly defined D-dimensional spatial indices to clarify the notation and facilitate the discussion of the interaction matrix.
- Section III, after Eq. (13): moved discussion of the explicit form of the matrix \( \mathcal{J} \) for the Ising model to a footnote; extended this discussion, corrected a previous typo and included a mention of the case of dimensions multiple of 4.
- Sections IV (bottom-left of p. 5) & V (top-right of p. 6): added to the main text the missing physical parameters relative to figures.
- End of Section IV: clarified that for small \(\Gamma \) a classical limit is approached.
- End of Section IV: added comments on the range of applicability of the approach and the regimes where it can be expected to work best.
- Throughout the manuscript: changed the nomenclature “Euclidean time” to “imaginary time”.

---

## Editorial Decision

published